

# Mercury and trace metal wet deposition across five stations in Alaska: controlling factors, spatial patterns, and source regions

Christopher Pearson[1], Dean Howard[2], Christopher Moore[3,4], and Daniel Obrist[2,3]

[1]Division of Hydrologic Sciences, Desert Research Institute, Reno, NV, USA

[2]Departmnet of Environmental, Earth, and Atmospheric Sciences, University of Massachusetts-Lowell, Lowell, MA, USA

[3]Division of Atmospheric Sciences, Desert Research Institute, Reno, NV, USA

[4]Gas Technology Institute, Des Plaines, IL, USA

*Correspondence to*: Daniel_Obrist@uml.edu

**Abstract.** A total of 1,360 weeks of mercury (Hg) wet deposition data were collected by the State of Alaska Department of Environmental Conservation and the U.S. National Park Service, across five stations covering up to eight years. Here, we analyse concentration patterns, source regions, and seasonal and annual deposition loadings across these five sites in Alaska, along with auxiliary trace metals including Cr, Ni, As, and Pb.

We found that Hg concentrations in precipitation at the two northern-most stations, Nome (64.5° N) along the coast of the Bering Sea and the inland site of Gates of the Arctic (66.9° N), were significantly higher (average of 5.3 ng L$^{-1}$ and 5.5 ng L$^{-1}$, respectively) than those at the two lowest-latitude sites, Kodiak Island (57.7° N, 2.7 ng L$^{-1}$) and Glacier Bay (58.5° N, 2.6 ng L$^{-1}$). These differences were largely explained by different precipitation regimes, with higher amounts of precipitation at the lower latitude stations leading to dilution effects. Highest annual Hg deposition loads

were consistently observed at Kodiak Island (4.80 +/- 1.04 μg m$^{-2}$), while lowest annual deposition was at Gates of the Arctic (2.11 +/- 0.67 μg m$^{-2}$). Across all stations and collection years, annual precipitation overwhelmingly controlled annual Hg deposition, explaining 73% of the variability in observed annual Hg deposition. Our analyses further showed that annual Hg deposition loads across all five Alaska sites were consistently among the lowest in the United States, ranking in the lowest 1 to 5 percent of over 99 monitoring stations.

Detailed back trajectory analyses showed diffuse source regions for most Hg deposition sites, which were almost identical with precipitation origins, suggesting global or regional Hg sources. One notable exception was Nome where we found pronounced differences between precipitation and Hg source origins with increased Hg contributions from the western Pacific Ocean downwind of East Asia. Analysis of multiple trace elements from Dutch Harbor, Nome, and Kodiak Island showed generally higher trace metal concentrations at the northern station Nome compared to

Kodiak Island further to the south, with concentrations at Dutch Harbor falling in-between. Across all sites, we find two distinct groups of correlating elements: Cr and Ni and As and Pb. We attribute these associations to possibly different source origins, whereby sources of Ni and Cr may be derived from crustal (e.g., dust) sources while As and Pb may include long-range transport of anthropogenic pollution. Neither Hg nor any of the other trace elements analyzed, consistently associated with these two groups of elements, suggesting largely diffuse source origins.

Calculations of enrichment factors (i.e., elemental enrichment compared to the upper continental crust) show low enrichment for Cr and Ni which is in support of a predominantly crustal source. High enrichment factors for Pb and



Se are indicative of anthropogenic or additional natural sources for these elements. For most other elements including Hg, enrichment factors fell in-between these groups showing no clear source attribution to either crustal or anthropogenic source origins.


## 1. Introduction

The land surface area of the State of Alaska is approximately one-fifth of that of the contiguous United States, but little spatial information is available on pollutant deposition and impacts affecting local ecosystems, wildlife, and humans. Most contaminant studies have focused on the Arctic domain, including studies conducted by large

international collaborative efforts such as the Arctic Monitoring and Assessment Program (AMAP), a working group of the Arctic Council (AMAP, 2009a, b, 2011). The interests in northern latitude pollutant studies are driven by reports that show significant neurotoxicity as well as immunological, cardiovascular, and reproductive effects in Arctic populations and wildlife from exposure to contaminants (AMAP, 2011). Important pollutants found in northern arctic and boreal areas include mercury (Hg) – which is the focus of the current analysis – as well as persistent organic

pollutant (POPs), polycyclic aromatic hydrocarbons (PAHs) and other trace metals such as lead (Pb) and cadmium (Cd) that are primarily supplied to the region by atmospheric transport and deposition (AMAP, 2005, 2009b, 2011). The major concerns of these pollutants are their toxicity and persistency in the environment. Delivery of contaminants to the high Arctic is expected to increase in the future due to changes in synoptic atmospheric transport patterns and an expected increase in contaminant source fluxes related to increased development, resource extraction, and

transportation activities within northern regions (Jaeglé, 2010;Streets et al., 2011).

Mercury is a neurotoxic pollutant significantly affecting northern latitudes, with human exposure mainly derived from consumption of seafood and marine mammals that are part of traditional diets based on hunting and fishing (Stow et al., 2011). Risks associated with long-term exposures to Hg, particularly to the organic monomethyl-Hg (MeHg), include neuro-developmental delays in children exposed in utero, impaired cardiovascular health in adults, and

disruption of immunological and endocrine functions (Karagas et al., 2012;Tan et al., 2009). Hg biomagnifies in aquatic and terrestrial food webs and is present at elevated concentrations in northern wildlife such as seals, polar bears, beluga whales, Arctic foxes, birds, and fish (Lawson and Mason, 1998;Watras and Bloom, 1992;Baeyens et al., 2003;Loseto et al., 2008;Evans et al., 2005;Dietz et al., 2009;Walker et al., 2006;Outridge et al., 2008;Douglas et al., 2012;Macdonald and Bewers, 1996;Leitch et al., 2007;Braune et al., 2014;Bocharova et al., 2013;Laird et al.,

2013;Ackerman et al., 2016;Eagles-Smith et al., 2016). Long-range transport via the atmosphere is considered the primary source of Hg deposition Alaska (Dommergue et al., 2010;Steffen et al., 2008). In addition, springtime photochemical reactions, termed Atmospheric Mercury Depletion Events (AMDEs), lead to additional Hg deposition to snow and ice, particularly along the Arctic Ocean coast (Douglas and Sturm, 2004;Lindberg et al., 2002). Regional and local sources of atmospheric Hg exist in Alaska both from natural and anthropogenic emissions. The EPA Toxics

Release Inventory (TRI) reported total air emissions of Hg and Hg compounds in Alaska of 24 kg (53 lbs) from both fugitive and point-source air sources in 2014 (Table 1; EPA, 2014). Other sources include natural emissions from wildfires (Mitchell et al., 2012;Wiedinmyer et al., 2006;Turetsky et al., 2006;Friedli et al., 2001;Brunke et al., 2001;Webster et al., 2016;Obrist et al., 2008), volcanic emissions (Mather and Pyle, 2004;Pyle and Mather,



2003;Nriagu and Becker, 2003;Ferrara et al., 2000), and degassing from Hg-enriched soils and possibly background

soils (Agnan et al., 2015a;Gustin et al., 2008), although many of these sources are poorly constrained across Alaska. Jaeglé (2010) conducted a detailed study on atmospheric sources of Hg contamination over Alaska using GEOS-Chem model simulations and suggested that anthropogenic emissions contribute approximately 57% of Hg deposition over Alaska, with other sources dominated by natural land (i.e. volcanos, wild fires) and ocean-based emissions.

Here we analyze Alaska Hg wet deposition data collected by the State of Alaska and the National Park Service between

2007 and 2015 across five Hg wet deposition stations in Alaska. Deposition sites are Gates of the Arctic, Nome, Glacier Bay National Park, Kodiak, and Dutch Harbor (**Table 1**; Please refer to Figure 4 for a map of site locations). The dataset contains 1360 weeks of total measurements across the five stations, with the longest record lasting almost 8 years, allowing for analysis of temporal, seasonal, and spatial patterns of Hg wet deposition across Alaska. All measurements were conducted according to trace metal sampling protocols following the National Atmospheric

Deposition Programs (NADP) Mercury Deposition Network (MDN) standards. We use statistical tests to compare deposition concentrations, loads, and seasonal and inter-annual patterns to assess variables controlling Hg deposition. In addition, we performed detailed back trajectory analyses for full annual datasets at select stations to quantify source regions that contribute to annual Hg deposition loads. We further use deposition data of auxiliary trace metals, including Cr, Ni, As, and Pb, to identify associations with Hg and among these trace elements and to derive enrichment

factors and source patterns. Finally, we performed spatial scaling and mapping of annual wet deposition of Hg throughout all of Alaska based on observed concentration gradients and precipitation distributions.

## 2. Materials and Methods

### 2.1. Collection and analysis of Hg Deposition data and ten additional trace elements

Weekly Hg deposition data were collected from five wet deposition stations in Alaska operated by the State of Alaska

Division of Environmental Conservation and the National Park Service. Deposition stations include Gates of the Arctic, Nome, Glacier Bay National Park, Kodiak, and Dutch Harbor (**Table 1**; Please refer to Figure 4 for a map of site locations). Gates of the Arctic is a protected wilderness area in northern Alaska consisting of multiple mountain ranges and sparse boreal forests. Nome is at sea level located on the western coast of Alaska off the Bering Sea. Glacier Bay National Park is a coastal site located in the northern section of the Alaska panhandle. Kodiak is a mountainous

island located off the southern coast of Alaska. Finally, Dutch Harbor is located on Amaknak Island in the Aleutian Islands. Sample collections were performed on a weekly basis using trace-metal wet-deposition collectors (Model MDN 00-125-4; N-Con Inc., Crawford, GA, USA), following MDN protocols for collection of Hg in precipitation (Mercury Deposition Network: Field Methods, 2017). In summary, the protocols include weekly collection using a specially modified NADP sampler. Each collection bottle and sample train was acid-cleaned prior to deployment and

pre-charged with hydrochloric acid preservative. Analysis of samples for Hg were performed by Mercury Analytical Laboratory (HAL, Eurofins Frontier Global Sciences, Inc., Seattle, Washington, USA) according to EPA Methods 1669 and 1631.

MDN data coverage varies by site and ranged from September 2007 to September 2015. The longest dataset is from Kodiak (2007 to 2015), followed by Gates of the Arctic (2008 to 2015), and Dutch Harbor (2009 to 2015). Both

Glacier Bay (2010 to 2013) and Nome (2013 to 2015) had shorter datasets (less than 3 years). All sites have





intermittent data gaps ranging from weeks to months. The most significant data gaps occurred at Kodiak during summers of 2009 and 2010, and Dutch Harbor during 2010, 2013, and 2014.

Detailed quality assurance and control of data followed the protocols of the NADP MDN. Each sample was assigned a quality rating (A, B, or C) based on collector performance, sample quality, and analytical measurement excellence.

A ratings were assigned to samples of "highest quality" with no issues during collection or analysis, B ratings referred to data with minor problems, and C ratings referred to samples with significant defects. Samples with a rating of C were removed prior to our data analysis. During weeks with missing rain gage data, the measured volume of water collected in the sample bottle was used as the precipitation measurement. Trace samples (unmeasurable by the rain gage) were assigned a precipitation value of 3.23 mm (following NADP MDN protocol).

In addition to the Hg deposition data, we analysed corresponding data of additional trace elements collected at three of the five Hg deposition stations: Dutch Harbor, Kodiak Island, and Nome. Available analysis of trace elements includes the following 10 additional elements: arsenic (As), beryllium (Be, cadmium (Cd), chromium (Cr), copper (Cu), lead (Pb), nickel (Ni), Selenium (Se), and zinc (Zn). Trace metal analysis was performed by inductively coupled plasma mass spectrometry following EPA Method 200.8 (Brockhoff et al., 1999). Seven of these trace elements are

listed on EPA's list of hazardous air pollutants (EPA, 2016), including As, Be, Cd, Cr, Pb, Ni, and Se. In total, there were 132 trace element deposition samples. Dutch Harbor had a total of 24 samples collected from 9/24/20113 through 4/28/2015, Nome had a total of 42 samples between 10/30/2013 and 4/28/2015, and Kodiak Island had a total of 65 samples between 9/17/2013 and 4/28/2015. We assumed that large gaps in data coverage were mainly due to low sample volumes collected during deposition samples which often did not allow measurement of these trace metals. A

significant number of samples showed trace element concentrations below the analytical detection limit. The percentage of samples below detection limits were as follows, from highest to lowest percentage: Cd: (80%), Be (56%), and Ni (44%), As (36%), and Cr (34%). All other elements had observations below reported detection limits less than 6%. Be and Cd were not included in this current analysis due to their large percentage of missing values.

### 2.2. Statistical analyses, spatial interpolation, and mapping

All statistical analyses were performed using the statistical software program "R" (R-Core-Team, 2014). Outliers for both the Trace Metal Dataset and NADP Hg Deposition dataset were determined using the 1.5 x Interquartile Range (IQR) rule. Datasets were $Log_{10}$-transformed and back-transformed during outlier testing in order to account for the skewed right/log-normal distribution of the data. Summary statistics are shown throughout this study using both the total data set and data sets with outliers removed. The trace metal dataset contained several elements where significant

portions of the data fell below detection limits (BDL). For these data, maximum likelihood estimation (MLE) summary statistics were calculated using the NADA package (Lee, 2013) in addition to 1/2 MDL substitution. In general, MLE and non-parametric techniques are preferred to be used for non-detect values since replacement of data (e.g., using ½ detection limits) can be problematic (Helsel, 2012). In particular MLE has been shown to produce unbiased estimates of mean, median, and standard deviation without substitution for minimally censored data sets with n >50 (Helsel,

2012). A lognormal distribution was assumed for MLE estimation. When applicable, non-parametric ranked-based tests were performed on the trace metal dataset. Boxplots and scatterplots were made using the ggplot2 package (Wickham, 2009).



Analysis of Variance (ANOVA) and Analysis of Covariance (ANCOVA) were performed using the aov function in the R stats package (R-Core-Team, 2014). Testing was performed on Log10-transformed data using a type-III

ANVOA to account for unbalanced factor levels in the dataset. Trace metal ANOVAs were performed on the ½ MDL substituted datasets.

Trace metal correlations were assessed by the non-parametric Kendall Tau test to account for BDL substitution and tied rankings. Kendall's test does not require any assumptions about the underlying distribution of the dataset and is less sensitive to tied values in the ranking process. Correlation testing was performed using the psych package

(Revelle, 2014) and visualized with the corrplot package.

A principal component analysis (PCA) using the R package ade4 (Dray and Dufour, 2007) was performed on the ranked trace metal dataset. The trace metal PCA was performed on ranked data to account for BDL values and non-normal distributions. Season was defined as follows for both datasets: Spring (March, April, May), Summer (June, July, August), Fall (September, October, November), Winter (December, January, February).

Mapping and spatial interpolation and extrapolation were performed in ArcGIS®. Inverse distance weighting (n=5; p=0.5) was used to interpolate and extrapolate precipitation weighted mean concentrations (PWM) across Alaska. The processing extent was allowed to extend beyond the sampling region to cover the entire State of Alaska. Precipitation weighted mean Hg concentrations were used from NADP MDN annual estimates and averaged for all available years. MDN annual estimates were only available for years with i) Hg sampling covering ≥75% of the sampling period ii)

Hg measurements of >75% of annual precipitation events and iii) >90% coverage of total annual precipitation (either gage or sample bottle). The estimated PWM Hg concentrations were then combined with annual normal precipitation averaged for the period of 2007-2015 from the NCEP Climate Forecast System Version 2 (CFSv2) 6-hourly Products (Saha, 2011) to estimate deposition totals across the State of Alaska. Precipitation data was accessed and compiled using Google Earth Engine (Google Earth Engine Team, 2015).

**2.3. Back-trajectory analysis and modeling and determination of deposition source areas**

Backtrajectories were generated using the National Oceanic and Atmospheric Administration Air Resources Laboratory's Hybrid Single Particle Lagrangian Integrated Trajectory (HYSPLIT) Version 4 Model. The model and associated files were accessed under: http://ready.arl.noaa.gov/hyreg/HYSPLIT_pchysplit.php. The model was implemented with meteorological data from NOAA for use in the HYSPLIT model available under:

http://ready.arl.noaa.gov/archives.php. Data used for this project was the Global Data Assimilation System (GDAS) 0.5 degree meteorological data. We performed an intensive analysis of the backtrajectories analyses for each individual precipitation event for the entire year of 2014 for each station with available Hg wet deposition for that year (Nome, Gates of the Arctic and Kodiak Island). Individual precipitation events were considered when measurable precipitation data were present at the one-hour time resolution. If a storm lasted several hours without an interruption of storm

activity, the event was classified as a single precipitation event; if there was an interruption in measureable precipitation >2hr in duration, the storm was separated into two events. Using this method, the number of precipitation events identified for 2014 were 247 (Kodiak Island), 182 (Nome), and 148 (Gates of the Arctic).

A single air parcel trajectory was calculated for each precipitation event, with the end time initialized to coincide with the center of each precipitation period along with the end latitude and longitude set to each monitoring station. The



185 altitude at which all backtrajectories was initiated was set to 2,000 m a.s.l. The output of each HYSPLIT run (trajectory) represents the latitude, longitude and altitude of an air parcel with these final coordinates over the previous 10 days, with temporal resolution of 1 hour (i.e. 240 3D coordinates per trajectory).

In order to delineate source regions for seasonal and annual wet deposition for all stations, each backtrajectory was weighted according to its contribution to annual Hg deposition as a fraction of the total annual Hg deposition sum. In

190 other words, each of the 240 3D coordinates of each trajectory output was allocated a scalar value representing the measured Hg deposition value for that precipitation event. We then calculated the residence time of each weighed backtrajectory in 2° x 2° grid cells, with the fractional residence time in a particular grid cell based on the time of the entire trajectory. For example, if a 10-day backtrajectory spent 12 hours of its time in a specific grid cell, it received a weighting of 5% for that particular grid cell. The weighting of a backtrajectory residence time in a grid cell was then

195 combined with the weighting of that trajectory as a fraction of total annual deposition, so that both the contribution to annual deposition as well as the residence time in grid cells were fully weighted. Finally, the sum of all backtrajectory weights for each grid cell were summed up and represented as a normalized frequency for each grid cell. We performed this both for 2014 Hg wet deposition, whereby the weighting occurred as the contribution of each trajectory to annual Hg wet deposition, and 2014-15 trace metal deposition. Each weighted and normalized, 2° x 2° grid of values was

200 finally converted to a raster file using the Python computing language package GDAL (PythonSoftwareFoundation;GDAL, 2016) and mapped and visualized in ArcGIS®.

### 3. Results and Discussion

#### 3.1. Spatial, seasonal, and temporal patterns of Hg wet deposition concentrations

We first present an analysis of concentration measurements of Hg and additional trace elements collected across the

205 five deposition stations. Minimum Hg concentrations at all stations were equal to the detection limit of the analyses (0.3 ng $L^{-1}$), and maximum concentrations strongly varied among stations. By far the highest Hg concentrations were reported for Gates of the Arctic with concentrations of up to 396 ng $L^{-1}$. Our statistical analysis determined such high values as outliers, which is in agreement with other datasets that found background level concentrations generally ranging from 3 to 5 ng $L^{-1}$ (White et al., 2009) and considered elevated Hg concentrations (e.g., in central Illinois)

210 when concentrations ranged above 20 ng $L^{-1}$ to 40 ng $L^{-1}$ (Lynam et al., 2014). For further analysis, we eliminated outlier concentrations (>26.14 ng $L^{-1}$) which we determined by an outlier analysis following the IQR rule whereby values above and below 1.5 x IQR were removed. For consistency among the deposition stations, we selected a common outlier concentration across all stations, rather than delineate an outlier concentration for each station separately. This outlier correction removed 1.9% of all data (17 values in total). Median Hg wet deposition

215 concentrations measured across the five deposition stations were in the following order, from highest to lowest (**Table 2**): Gates of the Artic (3.6 ng $L^{-1}$) > Nome (3.5 ng $L^{-1}$) > Dutch Harbor (2.3 ng $L^{-1}$) > Kodiak Island and Glacier Bay (both 1.8 ng $L^{-1}$). The distribution of mean values followed the same general order (**Table 2**), although mean values were higher compared to median values due to skewed distribution in concentration data.

220 Highest wet Hg deposition concentrations (**Table 2**) were observed at the two northernmost sites Gates of the Arctic and Nome, with median values almost double and statistically higher compared to concentrations of the two lower



latitude stations (Kodiak Island and Glacier Bay). A third station located at lower latitudes, Dutch Harbor, the westernmost station located on the Aleutian Islands, was similar in Hg concentrations as the two northern station and showed statistically higher concentrations (+28%) compared to the other two lower-latitude stations.

We determined that the major reason for higher Hg concentrations at northern sites was a lower dilution (or "wash-out" effect) of Hg concentrations by smaller storm sizes (**Figure 1** and discussion below). It is well known that large precipitation events (i.e., bigger storms or increasing duration of storms) lead to lower Hg wet deposition concentrations compared to small events. This is due to initially higher scavenging of airborne Hg, in particular of particulate-bound Hg (HgP) or gaseous oxidized Hg (GOM) (wash-out effect: Poissant and Pilote, 1998;Ferrara et al.,

1986), and has been observed in many studies (Lamborg et al., 1995;Mason et al., 1997;Landis et al., 2002;Lyman and Gustin, 2008;Faïn et al., 2011). Such "washout" effects also occurs in individual storms during which Hg concentrations are highest at the beginning of an event and decreases over time (Glass and Sorensen, 1999;Ferrara et al., 1986). However, the washout effect cannot explain the higher Hg concentrations at Dutch Harbor which were similar to those at the more northern stations (see discussion below).

**Figure 1a** shows the presence of the "washout" effect evident by inverse linear regressions between storm sizes (total weekly precipitation amounts) and respective measured weekly wet deposition Hg concentrations. All five stations showed statistically significant inverse correlations between the two variables. The slopes of the linear regressions, using $log_{10}$-transformed Hg concentrations (ng L$^{-1}$) and $log_{10}$-transformed precipitation (mm), varied between -0.28 and -0.46, but were not statistically different between stations (based on ANCOVA analyses). Overall, weekly

precipitation totals explained 28% of the variability in Hg concentrations ($r^2$=0.28, p-value<0.01, all sites). The common relationship between wet deposition concentrations and precipitation among all stations was best described by the following inverse linear relationship:

$$log_{10}(Hg_{conc.}[ng\ L^{-1}]) = 0.844 - 0.347 \times log_{10}(Precip[mm]) \qquad (1)$$

Cumulative distribution of daily storm sizes (**Figure 1b**) show that higher precipitation amounts occurred at lower-

latitude stations and were driving factors leading to their lower wet deposition concentrations. For example, precipitation totals at Gates of the Arctic and Nome were three to five times lower compared to the three lower latitude sites. Similar to Hg concentrations, differences in precipitation totals were statistically significant between the northern and lower-latitude sites, but not between the two northern or among the three lower-latitude sites (based on post-hoc comparison tests, not shown). The figure highlights a dominance of small precipitation events at Gate of

the Arctic and Nome, where a high fraction of precipitation events were below 1 mm. In comparison, the three lower latitude sites experienced much higher fractions of daily storms, e.g., above 2 mm. We propose that the washout effect largely accounts for higher Hg deposition concentrations at the dryer, northern sites compared to the lower-latitude sites Glacier Bay and Kodiak Island. As mentioned, this analysis, however, fails to explain why Dutch Harbor showed similarly high levels as the northern, more mesic sites.


**Figure 2** shows a pronounced seasonality of Hg wet deposition concentrations across all stations, with the highest Hg concentrations in summers, followed by spring, winter, and fall. ANOVA analysis across all sites resulted in statistically significant seasonal effects (variable "season": P<0.01), and post-hoc comparisons showed that Hg





concentrations differed among all seasons. The ANOVA also showed that seasonal patterns were consistent among the five stations with no significant differences among stations (i.e., no statistically significant interaction of "Season" x "Station"). Hence, seasonal patterns were relatively consistent among the five stations with median concentrations following the order summer>spring>winter>fall, with one exception being Dutch Harbor where fall concentrations were slightly above those in winter (2.0 ng L$^{-1}$ versus 1.9 ng L$^{-1}$).

Such seasonal patterns have been attributed to enhanced summertime GOM concentrations due to increased photochemical formation of oxidized mercury in summer that leads to increased atmospheric scavenging and higher Hg concentrations in precipitation (Pirrone and Mason, 2009;Selin and Jacob, 2008). Yet, we propose that seasonal patterns may also be affected by storm sizes and dilution effects since precipitation amounts were generally lowest in summer and highest in fall and winter. We performed analysis on seasonal differences by "detrending" data for different storm sizes, i.e., adjusting Hg concentrations by deducting the linear trend of the washout effect (equation 1). ANOVA and post-hoc Bonferroni comparisons of detrended Hg concentrations, however, showed that differences among seasons persisted after correcting for different precipitation sizes per season, and that the order of Hg$_{corr}$ concentrations followed the same order as the untrended concentrations (summer > spring> fall/winter). Hence, we propose a combination of Hg oxidation processes (Pirrone and Mason, 2009;Selin and Jacob, 2008) along with precipitation sizes contributing to seasonal differences.

### 3.2. Spatial, seasonal, and annual patterns of Hg wet deposition loads

In order to calculate annual deposition loads, NADP MDN protocols require substantial data coverage and stringent completeness criteria. These include, that the percentage of valid Hg samples exceed 75%; the percentage for which precipitation amounts were available, either from the rain gage or from the sample volume, exceed 90%; and the percentage of total measured precipitation associated with valid samples exceed 75%. To calculate annual wet deposition loads, MDN protocol uses multiplication of precipitation-weighted annual Hg concentration by annual precipitation records for each station. Following these constraints, data coverage allowed for a total of 16 years of annual wet deposition estimates across the five stations (**Table 3**).

Annual Hg deposition values across the five stations averaged 3.55±1.48 μg m$^{-2}$, with a minimum of 1.94 μg m$^{-2}$ at Gates of the Arctic in 2012 and a maximum of 5.74 μg m$^{-2}$ at Kodiak Island in 2011. In spite of differences in temporal coverage of annual Hg deposition among stations, we observed consistent differences among sites. When data from multiple stations were available, the highest Hg deposition loads were always observed at Kodiak Island and lowest at Gates of the Arctic. Differences in wet Hg deposition loads between stations were large: for example, in the four years of corresponding data, Kodiak Island deposition exceeded that at Dutch Harbor by a factor of 2.6 (in 2009), 2.4 (in 2011), 2.0 (in 2012), and 2.6 (in 2014). Second highest deposition loads were consistently observed at Dutch Harbor, with loads that were slightly below those in Kodiak Island in the two years of corresponding measurements. Statistical tests showed that annual deposition was statistically different among stations, and a post-hoc Bonferroni comparison showed that this difference was driven largely by a statistically significant difference between the highest (Kodiak Island) and lowest (Gates of the Arctic) station (P<0.05). When using a statistical significance level of 10% as opposed to 5%, we also observed significant differences between Kodiak Island and Glacier Bay and Kodiak Island and Nome. Hence, we generally can summarize annual deposition loads as follows: highest deposition occurred at





Kodiak Island (4.80±1.04 µg m$^{-2}$), but was not statistically different from the second highest station Dutch Harbor (4.52±1.47 µg m$^{-2}$), but statistically different from all other stations. Lowest deposition was observed at Gates of the Arctic (2.11±0.67µg m$^{-2}$), and intermediate values were observed for Glacier Bay (3.00±0.14 µg m$^{-2}$) and Nome (2.34 µg m$^{-2}$).

Using all data and stations, we did not observe significant effects of year of collection among stations (P = 0.138). However, there was substantial inter-annual variability in Hg deposition loads at individual stations. For example, using the five years of measurements at Kodiak Island, values ranged from 3.14 µg m$^{-2}$ (in 2009) to 5.61 µg m$^{-2}$ (in 2013), or a factor of 1.8 difference and a coefficient of variation of 22% (CV: Stdev/mean). The inter-annual comparison of Gates of the Arctic showed values from 1.19 µg m$^{-2}$ (2009) to 3.00 µg m$^{-2}$ (2010), or a factor of 2.5
difference and a CV of 32% (**Table 3**). Although the temporal coverage was too low to delineate clear inter-annual trends, the available data record generally shows low deposition in 2009 when lowest deposition occurred both at Gates of the Arctic and Kodiak Island.  In the year 2011, generally high deposition was observed, with highest deposition among all years observed at Dutch Harbor and Kodiak Island; however, the order of years was not fully consistent among all stations.

### 3.2.1. Annual deposition loads and relationships to annual precipitation

In order to characterize what drives annual deposition loads, we analysed precipitation-weighted mean annual Hg concentrations (PWM Hg) and annual precipitation, the two factors, which together constitute the annual deposition load. **Figure3a** shows a scatter plot and linear regression between PWM Hg and precipitation amounts using data of all years and all stations (16 values), showing a strong linear relationship between PWM Hg and precipitation. The
regression slope explains 59% of the variability in PWM Hg, and a slope of -0.0189 suggests that with each 100 mm increase in annual precipitation, PWM Hg concentration decreased on average by 1.9 ng L$^{-1}$. The patterns support a strong dependence of Hg concentrations on precipitation patterns, in agreement with the weekly Hg concentration data showing strong dilution effects as discussed above. Yet, an important difference is that the annual relationships between PWM Hg concentrations and precipitation is strongly linear, compared to non-linear functions between
weekly Hg wet deposition concentrations and weekly precipitation (i.e., $\log_{10}$-$\log_{10}$ relationships).

**Figure 3a** also shows that the range of annual precipitation is larger than the range of PWM Hg. For example, annual total precipitation differed by almost a factor of 12 (lowest annual precipitation of 27 mm in 2013 at Gates of the Arctic and highest precipitation of 316 mm at Kodiak Island in 2014). PWM Hg differed by a factor of 7 with lowest concentration of 1.5 ng L$^{-1}$ in Glacier Bay in 2011 and highest concentrations of 10.0 ng L$^{-1}$at Gates of the Arctic in
2010. When eliminating one unusually high PWM Hg concentration at Gates of the Arctic in 2010, the spread in PWM Hg of the remaining 15 station years was further reduced to a factor of 4. This suggests that annual precipitation had much stronger control in modulating annual deposition loads compared to PWM Hg concentrations. We conclude that a major control in determining Hg wet deposition loads across Alaska is annual precipitation, which alone explains 71% of the variability in observed annual deposition loads.
Compared to annual deposition observed across the contiguous U.S. (CONUS maps found at http://nadp.sws.uiuc.edu/mdn/), Hg deposition in Alaska was extremely low. Hg deposition across 99 deposition sites



of the contiguous U.S. in 2014 averaged 9.7±3.9 µg m$^{-2}$, with a median value of 9.0 µg m$^{-2}$. Of the three Alaskan stations that allowed for calculation of annual Hg deposition loads in 2014, Gates of the Arctic: (2.0 µg m$^{-2}$), Kodiak Island (5.1 µg m$^{-2}$), and Nome (2.4 µg m$^{-2}$) all showed very low deposition values compared to the other 99 U.S.

stations. When comparing the multi-year average annual Hg deposition of the Alaska stations to deposition values across the contiguous U.S. in 2014, three Alaska stations (Nome: 2.3 µg m$^{-2}$, Glacier Bay: 3.0 µg m$^{-2}$, Gates of the Arctic: 2.1 µg m$^{-2}$) showed annual deposition below all of the lower 48 contiguous U.S. States in 2014 (lowest value of 3.1 µg m$^{-2}$ observed at CA94, Converse Flats San Bernardino). Only Dutch Harbor (4.5 µg m$^{-2}$) and Kodiak Island (4.8 µg m$^{-2}$) exceeded the lowest deposition loads of the lower 48 States, yet even these two stations fell below the 5th

percentile of annual deposition observed at the 99 lower 48 States in 2014 (5.3 µg m$^{-2}$).

Similarly, PWM Hg concentrations were very low in Alaska compared to the rest of the U.S., which for the year 2014 averaged 10.6±9.1 ng L$^{-1}$ with a median value of 8.9 ng L$^{-1}$. Three Alaskan stations showed annual PWM Hg concentrations below the minimum concentrations (3.0 ng L$^{-1}$) of any of the stations in the contiguous United States, including Dutch Harbor (2.9 ng L$^{-1}$), Glacier Bay (1.9 ng L$^{-1}$), and Kodiak Island (2.2. ng L$^{-1}$). Nome with an annual

PWM concentration of 6.2 ng L$^{-1}$ and Gates of the Arctic (6.0 ng L$^{-1}$) were below the 15th percentile of concentrations of the lower 48th States. We conclude that low deposition values observed across coastal regions in Alaska were driven largely by very low wet deposition concentrations below concentrations at any of the contiguous U.S. deposition stations. For the two northern stations, Gates of the Arctic and Nome, extremely low wet Hg deposition was driven by a combination of low deposition concentrations and very low annual precipitation. Overall, very low concentrations

and deposition totals were observed throughout Alaska, typical of very remote areas with few local or regional point-sources and representative of more large-scale global background circulation patterns.

### 3.2.2. Spatial scaling of Hg deposition to the entire State of Alaska

We used spatial interpolation and extrapolation techniques to create maps of deposition concentrations and deposition loads across the state of Alaska, following interpolation protocols described by the National Atmospheric Deposition

Program (NADP, 2016). Limitations of such deposition maps, as stated by the NADP network, include: that "stations and maps represent regional trends (rather than local sources); that uncertainty with maps varies geographically, have not been quantified, and high levels of uncertainty can occur due to topographic variability, near urban and industrial areas, and in regions isolated from deposition sites". The NADP network specifically cautions making decisions based on projected maps when no direct measurements are available. For estimation of spatial deposition maps (i.e., sum of

deposition, seasonal and annual deposition loads), we did not remove values associated with outlier Hg concentrations and we included all officially released annual MDN deposition data that were quality controlled following official MDN protocols. In any case, because outliers were always associated with very low precipitation amounts, removal of outlier Hg concentrations have extremely small impacts on annual deposition loads. Concentration maps shown in **Figure 4** are based on the inverse distance weighting interpolation method of average PWM Hg concentrations for

each station (**Figure 4a**), and as such represents different collection years and number of years for each station based on the available data set. For example, the maps are based on 2 years (2011 and 2012) of wet Hg deposition data for





Dutch Harbor, 1 year (2014) for Nome, two years (2011 and 2012) for Glacier Bay, five years (2009-2014 without 2013) for Gates of the Arctic, and six years (2008-2014 without 2010) for Kodiak Island.

The resulting Hg concentration maps (shown in **Figure 4**) present a coarse spatial representation of the concentrations

patterns (i.e. only five measurement stations) that relate to precipitation gradients, with the highest concentrations observed at the northern two stations that show low annual precipitation and small storm sizes. The use of inverse weighting procedures results in interpolation of Hg concentrations that are not fully in accordance with the observed relationships to precipitation patterns. For example, precipitation maps show strong gradients in annual precipitation from the southern coast of Alaska to inland and northern locations, and relatively consistently low precipitation values

across much of central, northern, and eastern Alaska. In contrast, the interpolated Hg concentration map shows that interior and eastern Alaskan concentrations follow north-to-south gradients between the lower-latitude and higher-latitude stations, but do not account for east-west gradients. While we could have used precipitation-based estimates of Hg concentrations across the State (based on strong relationships of PWM Hg and annual precipitation), we decided not to deviate from common NADP mapping procedures.

**Figure 4b** shows precipitation maps across Alaska based on precipitation data averaged for the years 2007 to 2015. The long-term NOAA precipitation maps show strong gradients from the southern coastal locations to interior and northern Alaska, with very strong precipitation changes within short distance (50-100 miles). The highest annual precipitation was observed along the southeastern and southcentral coasts, with maximum precipitation of approximately 610 cm yr$^{-1}$. High precipitation amounts in the range of 200 to 300 cm yr$^{-1}$ were also observed in Kodiak

Island and Bristol Bay and the Aleutian/Probilof Islands. Moderate precipitation was observed in the southcentral and southwestern region of Alaska, generally in the range of 100 to 200 cm yr$^{-1}$, whereby occasionally higher levels of precipitation were observed in the mountain regions due to orographic precipitation effects. In the interior and far north regions of Alaska, however, annual precipitation sums were low and generally below 100 cm yr$^{-1}$.

The resulting annual deposition maps, i.e., the product of annual Hg$_{pw}$ concentrations and precipitation, are shown in

**Figure 4c**. Based on this map, we projected distinct zones of highest Hg deposition in Alaska along the southern and southeastern coasts, with annual Hg deposition exceeding 20 µg m$^{-2}$ yr$^{-1}$. The zones of highest annual Hg deposition, based on the estimated map, however, were confined to narrow zones of approximately 50-100 miles inland. Similarly, high Hg deposition may have occurred in isolated mountain areas near the southern coast such as in the Alaskan Range. For example, in the Denali National Park Region, estimated Hg wet deposition of up to 15 µg m$^{-2}$ yr$^{-1}$ was in

a similar magnitude of the highest deposition amounts along the southern and southeastern coast. Lower Hg deposition amount were projected, and in fact observed, along the southwestern coastal region, including Kodiak Island and the western and eastern Aleutians. Here, estimated annual Hg deposition were in the range 5 to 10 µg m$^{-2}$ yr$^{-1}$. Our estimated deposition maps indicated that in much of the State of Alaska, in particular in the interior and far northern regions, Hg deposition was very low, with annual Hg deposition generally below 4 µg m$^{-2}$ yr$^{-1}$ and in many areas (e.g.,

north of the Brooks Range) only in the range of 1-2 µg m$^{-2}$ yr$^{-1}$.

As stated above, estimated maps of annual deposition need to be considered with caution as they are based on interpolation methods and include a variety of possible errors. Compared to measured deposition at the five stations, estimated deposition fell well within 10% of observations at Gates of the Arctic and Kodiak Island. At other stations,



we found larger discrepancies between observed and modelled deposition, and at Glacier Bay and Nome discrepancies

were over 100%. We attributed these larger biases to discrepancies in annual precipitation: for example, at Glacier Bay and Nome, the model strongly overestimated precipitation (by 90% and 117%, respectively) which accounted for the main part of the bias. Reasons for precipitation errors were mainly due to the large grid size of the modelled precipitation combined with strong coastal gradients. For example, the deposition station at Glacier Bay, which was situated close to Point Gustavus in the inner Bay about 50 km inland from the main coast, was located along a very

large precipitation gradient which was not appropriately resolved by the grid size of the precipitation maps. Another possible reason for differences between observed and predicted deposition may include issues of precipitation fetch during deposition measurements. Precipitation gages generally show a strong bias towards under-catch of precipitation caused by wind, even with precipitation gauges that are designed with wind protection (Savina et al., 2012;Yang et al., 2000). Snowfall, which accounts for a very important fraction of annual precipitation in this area, can lead to

under-catch ranging from 20 to 50% during windy conditions (Rasmussen et al., 2012).

### 3.2.3.   Back-trajectory determine source regions of Hg deposition

We performed comprehensive back-trajectory analyses for the year 2014 which represented a typical deposition year and included data from the station with highest (Kodiak Island), lowest deposition (Gates of the Arctic) and intermediate (Nome) deposition amounts. As described in the methods section, individual backtrajectory modelling

was performed for all individual precipitation events (total of 247 events for Kodiak Island, 182 events for Nome, and 148 events for Gates of the Arctic) and subsequently each deposition event was weighted by its contribution to annual deposition load. Finally, we summarized residence times of all backtrajectories for all 0.5 x 0.5 degree grid cells. **Figure 5** shows normalized backtrajectory frequency maps for annual precipitation (panels a to c) and Hg deposition (panels d to f) for the year 2014.

For Kodiak Island, trajectory frequency maps showed almost identical patterns for precipitation and Hg deposition, both indicating the highest trajectory frequencies in close vicinity of the deposition station and to the south of the station. These patterns were attributed to the fact that each trajectory passed through adjacent station grid cells prior to arriving at the deposition station so that the vicinity of the stations always showed high contributions (both for precipitation and Hg deposition). In addition, major source origins for both precipitation and Hg wet deposition

stemmed from the Gulf of Alaska with additional contributions further south in the eastern Pacific Ocean up to a distance of 2,500 km south of Kodiak Island. A similar pattern was observed for Gates of the Arctic where close agreements existed between source origins of precipitation and Hg deposition. High contributions to annual Hg deposition and precipitation were observed again in the vicinity of the station, with additional source regions from the center of the Bering Sea and the Gulf of Alaska. There were only a few occasions where storms or deposition events

were tracked far into the western Pacific. We conclude that Gates of the Arctic experienced similar source regions for precipitation and Hg deposition and that these were predominantly located in the Bering Sea and the Gulf of Alaska. A different pattern was evident for Nome. Here, the frequency distributions of trajectories differed between precipitation and Hg wet deposition. Precipitation showed high source regions in the vicinity of the station and also relatively wide distribution across the Bering Sea, the central Pacific, and the western Pacific. For Hg, increased





contributions, relative to that of precipitation, were clearly visible in the western Pacific downwind of East Asia. This pattern indicated significant contributions from east Asia where known high Hg emission sources such as mining, industrial emissions, and coal burning have led to increased atmospheric Hg levels (Wong et al., 2006). A recent study by Pacyna et al. (2016), identified east Asia and India as the dominant source areas of global anthropogenic Hg emission from 2005 to 2010. Evidence that Hg pollution in East Asia contributes to elevated wet deposition Hg levels in downwind areas are also seen by the recently established Asia-Pacific Mercury Monitoring Network (APMMN) where preliminary data shows average wet deposition concentrations ranging from 7 to 23 ng $L^{-1}$ in samples covering areas from Vietnam to Korea (Sheu, 2017).

### 3.3. Auxiliary trace metal concentrations at Dutch Harbor (AK00), Kodiak Island (AK98), and Nome (AK04)

Across the three stations with data and deposition samples, we found the following order of median concentrations (MLE-based) of trace elements (**Table 4**): Zn (1.40 µg $L^{-1}$) > As (0.19 µg $L^{-1}$) > Cu (0.14 µg $L^{-1}$) > Se (0.06 µg $L^{-1}$) > Ni (0.04 µg $L^{-1}$) > Pb (0.04 µg $L^{-1}$) > Cr (0.02 µg $L^{-1}$) > Hg (0.002 µg $L^{-1}$). Highest concentrations were always observed for Zn which exceeded concentrations of all other elements by over an order of magnitude. Similarly, by far the lowest concentrations were always observed for Hg which was below concentrations of all other trace metals by at least one order of magnitude. Similar patterns of trace element concentrations, although generally higher in concentrations, have been observed in snow samples at lower latitudes, such as in Utah snowpack where Carling et al. (2012) observed highest bulk (unfiltered) concentrations of Zn in the range of 3-4 µg $L^{-1}$, with concentrations that exceeded that of other trace metals several-fold, and similarly, Hg concentrations were about one order of magnitude below concentrations of other trace elements. In the Everest region in the Himalayas, Lee et al. (2008) observed high concentrations of Zn (0.48 µg $L^{-1}$) as well compared to other trace elements (e.g., 0.11 for Cr, 0.08 for Pb and Ni, <0.01 for As). Here, concentrations were in a similar range as those observed in Alaska. In fresh snow in the French Alps, Veysseyre et al. (2001) observed concentrations of Zn up to 0.75 µg $L^{-1}$, again the highest compared to other trace metals (e.g., up to Cu: 0.2 µg $L^{-1}$) although Pb showed some high values in that study as well (max. of 1.76 µg $L^{-1}$).

An outlier analysis using the 1.5x IQR rule identified only a few points as outliers, i.e., only 1 or 2 for Se, Cr, Zn, Hg to a maximum of 6 for As. Similar to Hg, outlier concentrations were observed mainly at very low precipitation amounts, suggesting quality issues when low amounts of wet deposition were collected. For example, median precipitation of outlier data (all elements) was 0.46 cm, while the median precipitation of all samples was 2.7 cm. Therefore, we find evidence that at low precipitation, quality issues exist with trace metal samples, possibly due to blank values that result in unusually high concentrations at low precipitation. For further analysis and statistics, we therefore chose to remove outlier concentrations as many statistical tests are sensitive to such outliers.

### 3.3.1. Principal component analyses of the full trace element concentration dataset

In this section, we use Principal Component Analysis (PCAs) to explore commonalities of trace metals using both the entire dataset available across the three stations as well as individual stations. According to Reimann et al. (2008),





large geochemical datasets can use PCA to graphically inspect and reduce the data into a few components that may explain a high amount of the variability of the complete data.

**Figure 6** shows PCAs using all elements for all sites (panel a) and for each individual site separately (panels b-d), with a graphical representation showing the two main components (first component: x-axis; second component: y-axis). Similar patterns appeared in both the all data PCA and the individual site analyses. All elements showed a strong negative correlation with component one, suggesting that all element concentrations increased and decreased together. Interestingly, Hg showed the weakest correlation with component one, possibly related to Hg's highly volatile nature relative to other trace metals.

Ni, Cr, and Hg consistently fell on the negative side of the second component, while Pb and As fell on the positive side of the second component. Weaker associations were observed for Se, Pb, and As. Ni and Cr were likely associated because of a common source profile, possibly linked to crustal and/or natural sources (Carling et al., 2012;Agnan et al., 2015b;Veysseyre et al., 2001). On the opposite loading of this second principal component we found Pb and As, possibly due to their different source origins. As and Pb are primarily driven by anthropogenic, industrial emission sources such as smelters and combustion processes (Tchounwou et al., 2012), and these may in some part be derived from long-range transport from Asia. Hg's association with Ni and Cr supports a more background/natural source rather than local or point driven pollution source.

It is possible, however, that a portion of the factor loadings are related to site differences, as we did not find fully consistent patterns of elements across different sites. As discussed above, we observed that (i) all elements were statistically higher in concentrations at Nome compared to Kodiak Island; (ii) all elements except Pb and Se were statistically higher at Nome compared to Dutch Harbor; and (iii) only As, Pb, and Se were statistically higher at Dutch Harbor compared to Kodiak Island. It was therefore possible that factor 2 may reflect a different spatial distribution of As, Pb, and Se compared to most other elements (or particularly to Ni and Cr). Yet, analysis at individual sites (**Figure 6 panels b to d**) showed that the separation along component 2 was consistent across all sites. Hence, the pattern was consistent across all three sites with strong loading of Ni and Cr on one side of the second component and the opposite loading of As and Pb.

### 3.3.2. Enrichment factors of trace elements to assess geogenic (dust) and other sources including anthropogenic contributions

Further information about natural (geogenic/dust) versus anthropogenic sourcing were derived from calculations of enrichment factors. The calculation of enrichment factors of upper continental crustal distribution, ($EF_{ucc}$), showed ratios of elements of interest to a conservative crustal element such as Al. Al is a good tracer for crustal and rock elements and contributions from dust deposition. Using normalized ratios of other elements to that of upper continental crust (Wedepohl, 1995), the method calculates enrichments of elements above what is would be expected from purely natural sources. Enrichment factors were calculated following equation 3 (e.g., Carling et al., 2012):

$$EF_{ucc} = \frac{[X]_S/[Al]_S}{[X]_{ucc}/[Al]_{ucc}} \qquad\qquad (2)$$

Unfortunately, the dataset on Alaska wet deposition did not have data for Al nor other elements commonly used as conservative geogenic tracers such as Ti or Fe. In order to still perform calculations of enrichment factors, we decided



to use Cr and Ni instead as two possible reference elements. Both of these elements often show low enrichment factors compared to Al, indicating mainly crustal origins as for Al (Carling et al., 2012;Agnan et al., 2015b;Veysseyre et al., 2001). Uncertainties in this methodology include that local or regional soil elemental composition can be different from used reference crustal composition.

Calculated $EF_{ucc}$ are shown in **Figure 7** for all data from all three stations. $EF_{ucc}$ between 0.1 to 10 indicated that dominant sources were from soils, dust, or rocks; high $EF_{ucc}$ values were indicative of other natural or anthropogenic sources, whereby ratios between 10 to 500 were moderately enriched and values above 500 were strongly enriched and indicative of anthropogenic contributions  (Lee et al., 2008;Carling et al., 2012). Based on this, we would classify Cr, Pb, and Ni with median values below 10 as primarily derived from crustal contributions. The only element we

would clearly characterize as strongly enriched compared to crustal composition was Se with a median $EF_{ucc}$ of 1054. Moderately enriched $EF_{ucc}$ factors were observed for Cu, As, Zn, and Hg, with $EF_{ucc}$ values ranging from 11 to 57. **Figure 7** shows that the range and order of $EF_{ucc}$ was very consistent among the three stations, always showing a separation of element with distinctly different $EF_{ucc}$ values: low values for Cr, Pb, and Ni, median ranges for Cu, As, Zn, and Hg, and the highest value for Se.

This analysis suggested that the clustering of the elements Cr and Ni in the PCA analysis above was in fact likely driven by a common crustal origin. The results were also in support of the possibility that the opposite loading on the second principal component for Se and As could in part be driven by other sources (e.g., natural sources such as an ocean source for Se (Amouroux et al., 2001) or anthropogenic sources for As. For Pb, in contrast to PCA results, low $EF_{ucc}$ suggested crustal sources similar to Cr and Ni. We conclude that enrichment factors for Cr and Ni were showing

low enrichment factors in support of predominantly crustal sources, while high enrichment factors for Pb and Se suggested additional anthropogenic and natural sources. For most other elements, including Hg, enrichment factors were in-between not indicating clear crustal or anthropogenic sources.

## 4.   Conclusions

Our analysis of wet deposition data from five stations in Alaska found that Hg concentrations in precipitation at the

two northern stations (Nome and Gates of the Arctic) were consistently and significantly higher than the two lowest-latitude sites (Kodiak Island and Glacier Bay). These differences were largely explained by different precipitation regimes, with high amounts of precipitation at the lower latitude stations leading to washout effects compared to dryer, northern deposition sites. Differences in Hg concentrations between sites still existed after the effects of precipitation differences were removed, although the influence of precipitation was strong. After the correction, Gates of the Arctic

(AK06) and Nome (AK04) still had the highest Hg concentrations in precipitation and Kodiak Island statistically still had the lowest Hg concentrations. This suggested that other factors contributed to higher Hg concentrations in wet deposition at these stations as well.

Highest annual Hg deposition loads were always observed at Kodiak Island (AK98), and lowest deposition loads were always observed at Gates of the Arctic (AK06), and these differences were substantial. For example, Kodiak Island

(AK98) exceeded deposition at Dutch Harbor (AK00) by a factor 2.64 (in 2009), 2.40 (in 2011), 2.10 (in 2012), and 2.55 (in 2014). These patterns also were explained to a large degree by precipitation differences. Across all stations and collection years, precipitation overwhelmingly controlled annual Hg deposition; annual precipitation alone



explained 73% of the variability in observed annual Hg deposition across all stations and monitoring years. In comparison to Hg deposition loads across the contiguous Unites States, our analyses revealed that annual Hg
deposition loads for all of Alaska were among the lowest anywhere in the United States falling into the 5th percentile of all observed annual deposition. Based on observations and spatial interpolations, we found distinct zones of highest Hg deposition in Alaska along the southern and southeastern coasts (confined to 50-100 miles inland), and similarly high Hg depositions in isolated mountain areas near the southern coast, due to orographic precipitation enhancement. Lower Hg deposition amounts were observed along the southwestern coastal region, including Kodiak Island and the
western and eastern Aleutians. For most of the state, particularly in the interior and far northern regions, Hg deposition were estimated to be very low.

Back trajectory analysis of 2014 deposition data showed almost identical source origins of precipitation and Hg wet deposition, suggesting that the origin of Hg deposition was closely related to the origin of precipitation at the Gates of the Arctic (AK06) and Kodiak Island (AK98). Conversely, origins of precipitation and Hg wet deposition at Nome
(AK04) were quite different; for Hg deposition, we found increased source contributions relative to that of precipitation in the western Pacific Ocean near the East Asian continent, which could be due to long-range transport from East Asia.

PCA analyses revealed two distinct associations of trace elements: Cr and Ni were clustered, and so were As and Pb, which were attributable to different source origins. Sources of Ni and Cr are often considered driven by crustal (e.g.,
dust) sources), while As and Pb are attributable to anthropogenic inputs, including by long-range transport from Asia. Mercury, nor any of the other trace elements analysed, did not consistently associate with any of these four elements, suggesting more diffuse and possibly different source origins for these elements. Calculations of enrichment factors (i.e., elemental enrichment compared to the upper continental crust) showed low enrichment factors for Cr and Ni in support of predominantly crustal sources, while high enrichment factors for Pb and Se suggested additional
anthropogenic and natural sources. For most other elements, including Hg, enrichment factors were in between these elements, not indicating a clear attribution to either crustal or anthropogenic source origins.

Based on our findings, we recommend continued monitoring of Hg wet deposition at select sites designed to capture the strong precipitation gradients observed throughout Alaska. Given the low wet deposition amounts of Hg across most of Alaska, we recommend additional focus on dry deposition monitoring, as recent research has suggested that
deposition of gaseous elemental Hg ($Hg^0$) dominates Hg loading in most terrestrial ecosystems. Additional collection and analysis of typical source tracers (e.g., Fe, Al, or Ti for dust; Cl, Na, Mg for ocean sources; gaseous tracers such as CO, $CO_2$, and $O_3$ as combustion tracers; $^{222}Rn$ as a boundary layer tracer) may help facilitate better source apportionment both for wet deposition as well as for gaseous species (e.g., $Hg^0$).

### 5. Author Contributions

CP and DO designed this analysis. CP analysed the data and generated figures. DH performed the HYSPLIT backtrajectory modelling. CP and DO prepared the manuscript with contributions from all co-authors.

### 6. Acknowledgements

This work was funded by the Alaska Department of Environmental Conservation Grant #: USFSW/15.668/F12AF00730. Funding was also provided by a collaborative research project by the U.S. National
Science Foundation (Award #1739567 and 1304305).





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

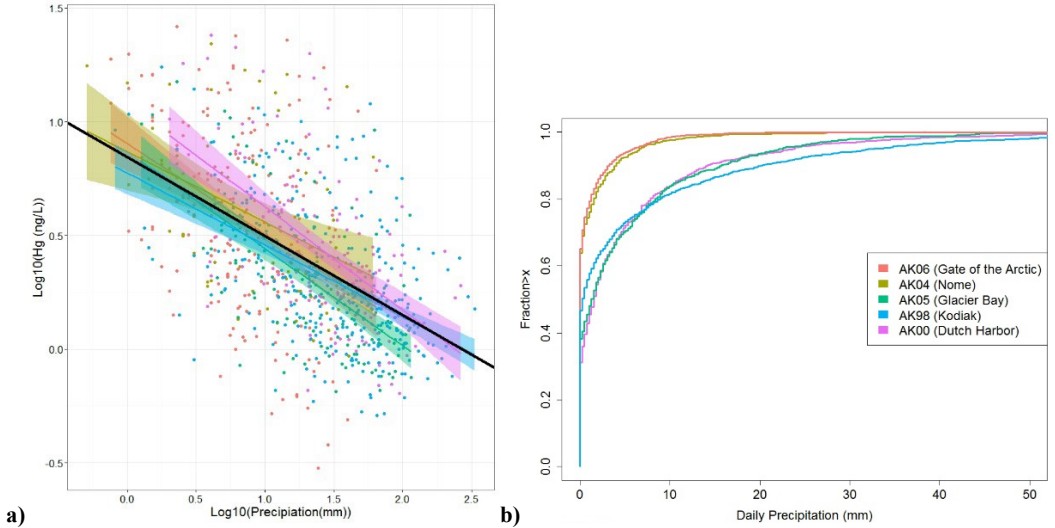

**a)**                                                                                 **b)**

**Figure 1. a) Scatterplots of observed Hg concentrations and precipitation amounts, separated by station. The black line shows the overall regression using all sites/all data. A clear dilution effect is observed at all sites. b) Empirical cumulative distribution plot of daily precipitation at five monitoring stations in Alaska (plot has been cropped at 50 mm). Distinct differences in storm size are observed between the northern (red and gold lines) and southern stations (green, blue, and**
**purple lines).**





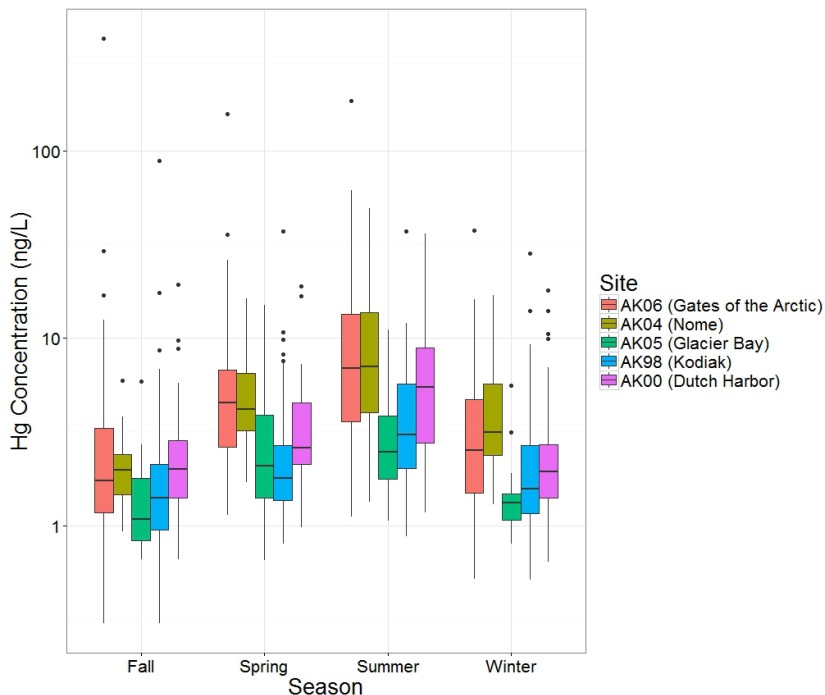

**Figure 2. Summary boxplot of Hg concentrations separated by monitoring station and collection season. Similar seasonal trends are observed at each site with the highest concentrations occurring in summer and lowest concentrations in fall.**

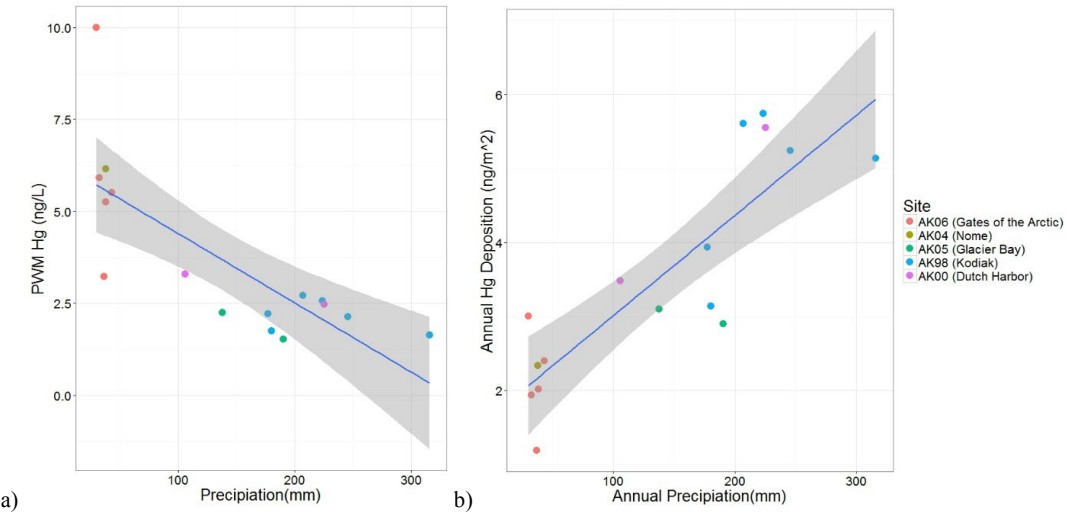

**Figure 3. a) Plot of annual precipitation-weighted mean (PWM) Hg concentration verses annual precipitation (mm). Linear regression analysis shows a significant relationship with a correlation coefficient $R^2$=0.557, p-value= <0.001. b) Plot of annual Hg deposition verses annual precipitation (mm). Linear regression analysis shows a significant relationship with a correlation coefficient of $R^2$=0.7141, p-value= <0.001.**

790          a)

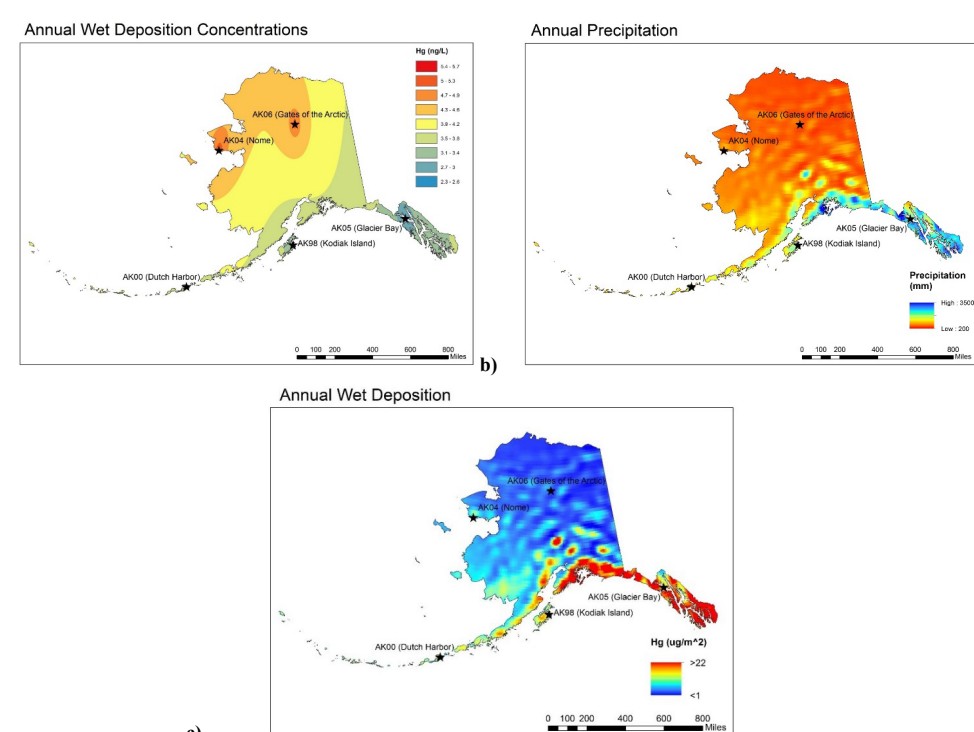

c)

**Figure 4. Average Annual Hg deposition maps for the state of Alaska:  a) Inverse distance weighted concentration layer; b) NCEP Climate Forecast System Precipitation 2007-2015 annual average; c) Hg deposition estimates. Maps show large-scale concentration gradients (north/south) largely due to precipitation differences with total loading being highly dependent on precipitation.**


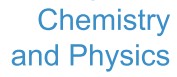
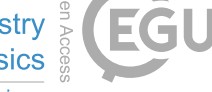


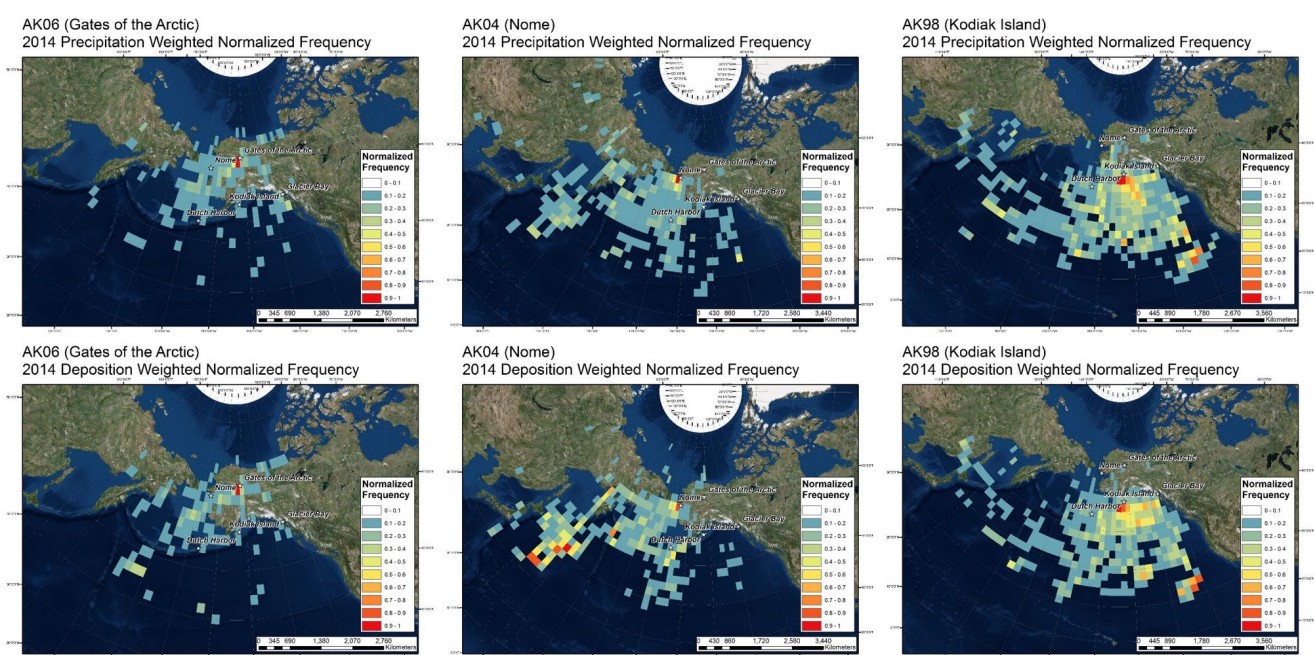

Figure 5. Normalized-frequency maps weighted by top) precipitation and bottom) Hg deposition for left) Gates of the Arctic, middle) Nome, and right) Kodiak Island.

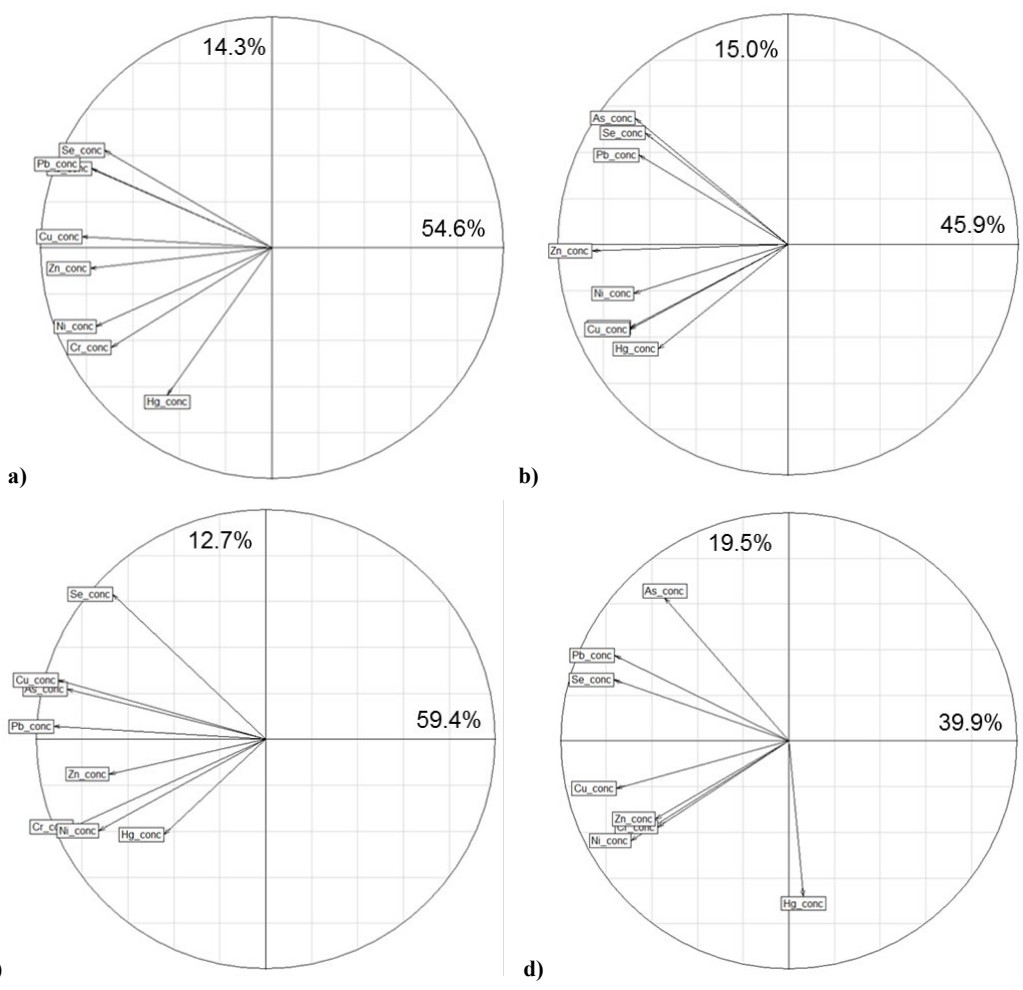

c)                                        d)

**Figure 6. Trace metal rank-based principal component analysis bipolots of component 1 (x-axis) versus component 2 (y-axis) for a) All Sites, b) Dutch Harbor, c) Nome, and d) Kodiak Island. Values represent the % variance explained by each relative component.**






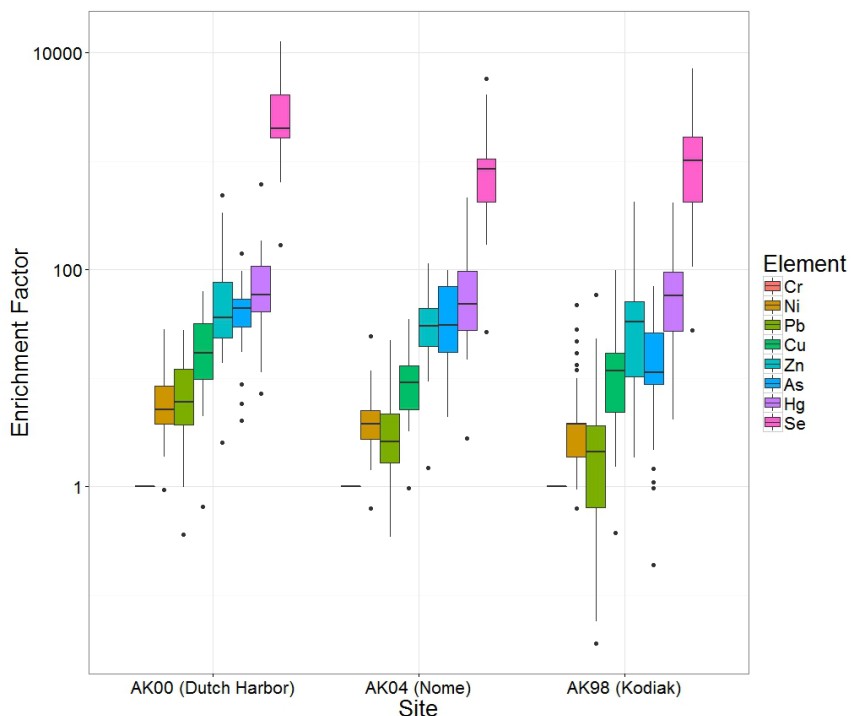

**Figure 7. Summary boxplot of enrichment factors separated by site. Enrichment factors are calculated using Chromium as the conservative tracer.**

**Table 1. Overview of available data coverage for each station.**

| Station ID | AK06 | AK04 | AK05 | AK98 | AK00 |
|---|---|---|---|---|---|
| Station Name | Gates of the Arctic National Park | Nome | Glacier Bay National Park | Kodiak | Dutch Harbor |
| Latitude | 66.906 | 64.506 | 58.457 | 57.719 | 53.845 |
| Longitude | -151.683 | -165.396 | -135.867 | -152.562 | -166.505 |
| Elevation (m) | 630 | 15 | 2 | 7 | 58 |
| Start Measurements | 11/11/2008 | 09/25/2013 | 03/16/2010 | 09/18/2007 | 09/26/2009 |
| Stop Measurements | 10/27/2015 | 09/29/2015 | 05/21/2013 | 09/29/2015 | 09/30/2015 |
| # of Weeks | 362 | 103 | 165 | 418 | 312 |
| # of Hg Concentrations (QR A and B) | 216 | 67 | 137 | 321 | 145 |
| # of Hg Depositions (QR A and B) | 267 | 85 | 144 | 351 | 152 |
| Data Coverage (Deposition) | 74% | 83% | 87% | 84% | 49% |





**Table 2. Summary statistics of Hg concentrations observed at the five deposition stations, and ANOVA analysis and post-hoc comparisons to test for statistical differences in Hg concentrations among different stations. ANOVA analyses were performed after removal of outliers (concentrations > 26.14 ng L$^{-1}$).**


| Station ID | AK06 | AK04 | AK05 | AK98 | AK00 |
|---|---|---|---|---|---|
| Station Name | Gates of the Arctic National Park | Nome | Glacier Bay National Park | Kodiak | Dutch Harbor |
| **Hg concentrations (ng L$^{-1}$)** | | | | | |
| # Outliers Removed (>26.14 ng L$^{-1}$) | 11 | 1 | 0 | 4 | 1 |
| Mean | 5.3 | 5.5 | 2.6 | 2.7 | 4.0 |
| Median | 3.6 | 3.5 | 1.8 | 1.8 | 2.3 |
| Standard Deviation | 4.9 | 5.0 | 2.5 | 2.4 | 4.4 |
| Minimum | 0.3 | 0.9 | 0.7 | 0.3 | 0.6 |
| Maximum | 26.1 | 22.0 | 15.0 | 17.4 | 24.0 |
| **ANOVA Results [Log10(Hg(ng L$^{-1}$))~Season*Site]** | | | | | |
| | Df | SS | RSS | AIC | F-value |
| Season | 3 | 90.91 | 502.35 | -440.70 | 62.46 |
| Site | 4 | 57.06 | 468.51 | -503.25 | 29.40 |
| Season: Site Interaction | 12 | 8.60 | 420.04 | -614.03 | 1.48 |
| **Post-Hoc Comparisons** | | | | | |
| **Season** | Diff | Lower | Upper | P-value | |
| Spring-Fall | 0.4722 | 0.3016 | 0.6429 | <0.001 | |
| Summer-Fall | 0.9155 | 0.7492 | 1.0818 | <0.001 | |
| Winter-Fall | 0.1919 | 0.0174 | 0.3664 | 0.0245 | |
| Summer-Spring | 0.4433 | 0.2725 | 0.6141 | <0.001 | |
| Winter-Spring | -0.2803 | -0.4591 | -0.1015 | <0.001 | |
| Winter-Summer | -0.7236 | -0.8983 | -0.5489 | <0.001 | |
| **Site** | Diff | Lower | Upper | P-value | |
| Dutch Harbor-Nome | 0.2498 | -0.0333 | 0.5328 | 0.1129 | |
| Dutch Harbor-Kodiak | -0.3614 | -0.5528 | -0.1701 | <0.001 | |
| Dutch Harbor-Glacier Bay | -0.4630 | -0.6903 | -0.2358 | <0.001 | |
| Dutch Harbor-Gates of the Arctic | 0.1261 | -0.0812 | 0.3333 | 0.4577 | |
| Nome-Kodiak | -0.6112 | -0.8689 | -0.3536 | <0.001 | |
| Nome-Glacier Bay | -0.7128 | -0.9981 | -0.4275 | <0.001 | |
| Nome-Gates of the Arctic | -0.1237 | -0.3934 | 0.1459 | 0.7193 | |
| Kodiak-Glacier Bay | -0.1016 | -0.2963 | 0.0931 | 0.6106 | |
| Kodiak-Gates of the Arctic | 0.4875 | 0.3166 | 0.6584 | <0.001 | |
| Glacier Bay-Gates of the Arctic | 0.5891 | 0.3788 | 0.7994 | <0.001 | |





**Table 3. Summary statistics of NADP MDN annual estimates of precipitation-weighted mean concentration, deposition, and precipitation for five monitoring sites in Alaska.**

| Station ID | AK06 | AK04 | AK05 | AK98 | AK00 |
|---|---|---|---|---|---|
| Station Name | Gates of the Arctic National Park | Nome | Glacier Bay National Park | Kodiak | Dutch Harbor |
| # of Years | 5 | 1 | 2 | 6 | 2 |
| *Hg PWM concentrations (ng L⁻¹)* | | | | | |
| Mean | 5.980 | 6.153 | 1.887 | 2.167 | 2.875 |
| Standard Deviation | 2.474 | | 0.515 | 0.431 | 0.581 |
| Median | 5.509 | 6.153 | 1.887 | 2.177 | 2.875 |
| Minimum | 3.224 | 6.153 | 1.523 | 1.628 | 2.464 |
| Maximum | 9.997 | 6.153 | 2.251 | 2.709 | 3.286 |
| *Hg deposition (ng m⁻²)* | | | | | |
| Mean | 2.108 | 2.338 | 3.002 | 4.801 | 4.518 |
| Standard Deviation | 0.665 | NA | 0.145 | 1.035 | 1.466 |
| Median | 2.018 | 2.338 | 3.002 | 5.191 | 4.518 |
| Minimum | 1.188 | 2.338 | 2.899 | 3.137 | 3.481 |
| Maximum | 3.004 | 2.338 | 3.104 | 5.743 | 5.554 |
| *Precipitation (mm)* | | | | | |
| Mean | 363.1 | 380.0 | 1641.7 | 2249.4 | 1657.4 |
| Standard Deviation | 52.1 | 0.0 | 370.6 | 515.3 | 845.2 |
| Median | 368.6 | 380.0 | 1641.7 | 2153.2 | 1657.4 |
| Minimum | 300.5 | 380.0 | 1379.6 | 1773.3 | 1059.7 |
| Maximum | 435.1 | 380.0 | 1903.8 | 3157.9 | 2255.1 |

**Table 4. Trace metal summary statistics using both Maximum Likelihood Estimation and ½ MDL substitution techniques.**

*Maximum Likelihood Summary Statistics (ug L⁻¹; n=131)*

| | n < MDL | median | mean | std deviation |
|---|---|---|---|---|
| Arsenic | 47 | 0.0216 | 0.1936 | 1.7240 |
| Chromium | 45 | 0.0222 | 0.0793 | 0.2719 |
| Copper | 1 | 0.1370 | 0.2525 | 0.3907 |
| Lead | 7 | 0.0389 | 0.1080 | 0.2799 |
| Nickel | 57 | 0.0360 | 0.2782 | 2.1331 |
| Selenium | 31 | 0.0635 | 0.1125 | 0.1646 |
| Zinc | 2 | 1.4033 | 2.9077 | 5.2769 |