# Peer review of "Mercury and trace metal wet deposition across five stations in Alaska: controlling factors, spatial patterns, and source regions"

_Atmospheric Chemistry and Physics, 2018_

## Referee Comment (RC1) · Anonymous Referee #1 · 29 Dec 2018

This manuscript provides a detailed analysis of long term atmospheric chemical measurements at sites in Alaska. The authors provide a wealth of statistical and back trajectory analyses and the wealth of information is well presented. This manuscript will be of interest to a variety of readers and is well suited for ACPD. There are a lot of small grammatical and typographical errors and it is frustrating to see this and have to address it all. In the future I recommend all authors read and edit and fix these issues as I do not feel it is a Reviewer's job to fix grammar and punctuation. That said I recommend minor editing and a few suggestions but overall I strongly recommend this for

publication.

General comments keyed to the text:

12: no comma after Service 13: after "years." I recommend a sentence identifying the locations of the stations. I also recommend mentioning here that there are data from a variety of other metals otherwise at line 31 we see mention of the other metals. 16: were statistically significantly? 30: here and elsewhere (line 38) it is "in between" with no hyphen 66: to Alaska 76: study of 78: Perhaps a sentence here providing context for Hg deposition attribution from other locations? Perhaps the Lower 48 since that is brought in later for the other metals. Is 57% high, low, or likely about average for global sources and deposition? 86: Program's 98-101: I like that a little description of the terrain and vegetation is given for the Gates of the Arctic site but what about the others? Add some more info please. Maybe a sentence for each site? 128-9: "due to the low sample volume collected for each deposition sample"? 220: The highest 231: also occur 232: and decrease 249: Gates 259-264: what about the typical and long term different fractions of wet and dry precipitation at each site? Is this changing over time? How were snow samples collected? And is there any sense that the dry precipitation is shifting towards wet? Particularly at the more northern sites? This could feed into some comments I have later about the future deposition.

Were there any major storm events that stood out in the analysis? I realize long precip event samples were broken up but can they be pieced back together to identify how/where large precip events may affect the overall yearly deposition at a site? I realize this may be a giant analysis that I do not want to send the authors out on but I am curious. This is sort of addressed in the next few lines.

280: the MDN 328-9: This is an extremely important finding. Figure 4a: why do Gates of the Arctic and Nome have seemingly anomalously higher values (ie the small circles of higher color keyed values) only where the stations are located? I assume some sort of kriging of data analysis artefact?

[Figure]

394: Since Denali National Park is mentioned. Isn't there some data or results from Denali? Again I do not want the authors to spend a lot of time on this but I wonder if there are similar results or analyses from any other locations. Or other studies with similar approach applied to Alaska that could be referenced? What about the long term DOE air monitoring sites- do they measure metals or Hg?

395: amounts 401: maps of estimated 418: remove "deposition" after "lowest" 419-420: "individual used twice. Can this be cleaned up to one mention of "individual"?

435-445: This is an interesting result of the study.

523: elements 525: suggests. Here and elsewhere I recommend active and not passive tense. 526: the results also support the possibility 530: crustal sources while (no comma) 533: in between and thereby do not indicate (if you agree to shift to active tense)

General comment: There are an increasing number of studies showing that the Arctic is getting wetter, particularly that the winter is shortening and the snow to rain fraction is decreasing. Could the authors break their data into snow versus rain as the seasonal sources and then use potential projections to address who/where a wetter Arctic may affect deposition? At the least there should be some mention of how a warmer future Arctic and its' changing precipitation dynamic may affect loadings.

Figure 2: how were the different season parsed? And were snow versus rain events separated? I realize the coastal sites may see lots of winter rain but I am curious again at the snow versus rain breakdowns.

Figure 5: The areas projected by the true color images (ie the map area) are slightly different. I recommend providing one consistent background and maps at the same scale to show the different source regions and distances of back trajectories.
* * *

---

## Referee Comment (RC2) · Anonymous Referee #2 · 2 Jan 2019

Here are a few concerns this reviewer has. One problem is their analysis of "precipitation origins/sources". Precipitation formation is in large part driven by microphysics; it is not like pollutants that can be transported from upwind source regions. This sort of analysis and language is really odd. Hence quite a bit of their "intensive" trajectory analysis for "precipitation origins/sources" is not valid.

Section 2.2 is not necessary. The maps were extrapolated from 5 sites only apparently with too large uncertainties. For instance, the spatial distributions of Hg concentrations can be quite complex. Their results showed that Dutch Harbor saw a similar precipitation amount to that at Glacier but had a median Hg concentration >50% greater, and

[Figure]

Kodiak had >30% more precipitation and >15% more Hg concentration that those at Glacier. How to reconcile these disparate spatial differences? Why would one expect simplistic linear extrapolation to capture these differences?

Section 3.3 needs to be redone. First the authors conducted a PCA analysis. Usually one uses tracers that represent distinctly different sources, but the metals they used could not seem to do the job. It was not clear why they did a PCA of the metals to begin with. Since Al measurements were not available, the authors decided that Cr and Ni can be used as alternatives of Al, a crustal tracer. Where did they get [xucc]/[Crucc] or [xucc]/[Niucc]? At one point the authors decided that Asian pollution could influence Alaska based on As and Pb (line 489). But there are major anthropogenic sources for Cr and Ni in Asia as well. It was unclear what the authors were trying to do with Figure 6. No interpretation was given but merely description of how the first two components were positively or negatively correlated. What do those correlations really mean?

This manuscript can be shortened significantly, by removing tutorial material, redundancy, repetition, and passages that merely pointed out the obvious. To be specific, Section 2 can be cut down quite a bit by removing the tutorial stuff in the statistics and trajectory sections. In their results and discussion sections they often stated the obvious.

---

## Author Comment (AC1) · 14 Mar 2019

This manuscript provides a detailed analysis of long term atmospheric chemical measurements at sites in Alaska. The authors provide a wealth of statistical and back trajectory analyses and the wealth of information is well presented. This manuscript will be of interest to a variety of readers and is well suited for ACPD. There are a lot of small grammatical and typographical errors and it is frustrating to see this and have to address it all. In the future I recommend all authors read and edit and fix these issues as I do not feel it is a Reviewer's job to fix grammar and punctuation. That said I recommend minor editing and a few suggestions but overall I strongly recommend this for publication.

General comments keyed to the text:
12: no comma after Service
*Removed comma.*

13: after "years." I recommend a sentence identifying the locations of the stations. I also recommend mentioning here that there are data from a variety of other metals otherwise at line 31 we see mention of the other metals.
*We believe we already appropriately give the location of the five stations in the abstract, and highlight in line 14 that additional metals (Cr, Ni, As, and Pb) were analysed.*

16: were statistically significantly?
 *Modified wording to "statistically higher"*

30: here and elsewhere (line 38) it is "in between" with no hyphen
*Edited to consistently use "in-between" throughout paper.*

66: to Alaska
*Corrected.*

76: study of
*Corrected.*

78: Perhaps a sentence here providing context for Hg deposition attribution from other locations? Perhaps the Lower 48 since that is brought in later for the other metals. Is 57% high, low, or likely about average for global sources and deposition?
*Added citation for context:*
*This estimate may be high given that globally, anthropogenic Hg emissions are estimated to account for approximately 30% of total atmospheric sources (i.e., total anthropogenic and natural emissions plus re-emission) (UNEP 2013).*

86: Program's
*Corrected.*

98-101: I like that a little description of the terrain and vegetation is given for the Gates of the Arctic site but what about the others? Add some more info please. Maybe a sentence for each site?
*Added additional vegetation and landcover descriptions for each site.*

128-9: "due to the low sample volume collected for each deposition sample"?
*Edited to "due to low sample volumes collected during sampling"*

220: The highest
*Corrected.*

231: also occur
*Corrected.*

232: and decrease
*Corrected.*

249: Gates
*Corrected.*

259-264: what about the typical and long term different fractions of wet and dry precipitation at each site? Is this changing over time? How were snow samples collected? And is there any sense that the dry precipitation is shifting towards wet? Particularly at the more northern sites? This could feed into some comments I have later about the future deposition. Were there any major storm events that stood out in the analysis? I realize long precip event samples were broken up but can they be pieced back together to identify how/where large precip events may affect the overall yearly deposition at a site? I realize this may be a giant analysis that I do not want to send the authors out on but I am curious. This is sort of addressed in the next few lines.

*We clarify that all samples were collected using the standard NADP wet deposition sampling protocols which do not analyze individual storms based on 2-week sampling periods in the NADP program. However, we already discuss the close relationships between deposition concentrations/amounts and precipitation size in detail in section 3.1. (*We determined that the major reason for higher Hg concentrations at northern sites was a lower dilution (or "wash-out" effect) of Hg concentrations by smaller storm sizes (**Figure 1** and discussion below)….and following paragraph.)

*The reviewer is correct about the importance of dry deposition. We added a short section in paragraph 3.2.1. about the importance of dry deposition that is based on recent studies in the Arctic tundra.*

280: the MDN
*Corrected.*

328-9: This is an extremely important finding. Figure 4a: why do Gates of the Arctic and Nome have seemingly anomalously higher values (ie the small circles of higher color keyed values) only where the stations are located? I assume some sort of kriging of data analysis artefact?
*The "hotspots" of seemingly higher values at Gates of the Arctic and Nome are due to limitations with IDW interpolation and a small number of sites. We applied IDW to follow standard methods developed by NADP utilized for CONUS deposition mapping. In general, the concentration map agrees with the precipitation map and shows higher concentration in the dryer northern portions of AK and lower concentrations at the wetter southern and coastal sites. We caution readers about the limitations of applying spatial interpolation with such a limited number of sites, but felt that the overall figure demonstrates the spatial patterns found between these five sites.*

394: Since Denali National Park is mentioned. Isn't there some data or results from Denali? Again I do not want the authors to spend a lot of time on this but I wonder if there are similar results or analyses from any other locations. Or other studies with similar approach applied to Alaska that could be referenced? What about the long term DOE air monitoring sites- do they measure metals or Hg?
*To our knowledge, there are no published wet deposition studies of Hg in Alaska, with the exception of the study by Jaeglé we cite in the manuscript.*

395: amounts
*Corrected.*

401: maps of estimated
*Corrected.*

418: remove "deposition" after "lowest"
*Corrected.*

419-420: "individual used twice. Can this be cleaned up to one mention of "individual"?
*Corrected.*

435-445: This is an interesting result of the study.

523: elements
*Corrected.*

525: suggests. Here and elsewhere I recommend active and not passive tense.
*Corrected here and throughout.*

526: the results also support the possibility
*Corrected.*

530: crustal sources while (no comma)
*Corrected.*

533: in between and thereby do not indicate (if you agree to shift to active tense)
*Corrected.*

General comment: There are an increasing number of studies showing that the Arctic is getting wetter, particularly that the winter is shortening and the snow to rain fraction is decreasing. Could the authors break their data into snow versus rain as the seasonal sources and then use potential projections to address who/where a wetter Arctic may affect deposition? At the least there should be some mention of how a warmer future Arctic and its' changing precipitation dynamic may affect loadings.
*We added a few sentences in the summary section discussing impatct of global warming and arctic amplification on Hg deposition and other relevant ecosystem processes. While we cannot discuss specific responses on Hg deposition, we highlight global warming will result in complex, yet poorly understood, consequences of climate change on Arctic Hg exposure.*

Figure 2: how were the different season parsed? And were snow versus rain events separated? I realize the coastal sites may see lots of winter rain but I am curious again at the snow versus rain breakdowns.

*We added season definitions to Figure text for clarity.*

Figure 5: The areas projected by the true color images (ie the map area) are slightly different. I recommend providing one consistent background and maps at the same scale to show the different source regions and distances of back trajectories.

*Figure was regenerated to use same spatial extent for all plots.*

---

## Author Comment (AC2)

Here are a few concerns this reviewer has. One problem is their analysis of "precipitation origins/sources". Precipitation formation is in large part driven by microphysics; it is not like pollutants that can be transported from upwind source regions. This sort of analysis and language is really odd. Hence quite a bit of their "intensive" trajectory analysis for "precipitation origins/sources" is not valid.

*This is a good point. We removed the trajectory analyses for precipitation origins/sources, and now only present the analysis for Hg deposition. We also adjusted the text accordingly.*

Section 2.2 is not necessary.
*We disagree (see also comments below). While we agree that there are in fact only five stations, we strongly feel that providing these maps ideally highlights that large parts of this northern parts is expected to have very low Hg deposition loads due to low precipitation amounts. We provided sufficient caution to read these maps in the text and highlight the limitations of the maps as well.*

The maps were extrapolated from 5 sites only apparently with too large uncertainties. For instance, the spatial distributions of Hg concentrations can be quite complex. Their results showed that Dutch Harbor saw a similar precipitation amount to that at Glacier but had a median Hg concentration >50% greater, and Kodiak had >30% more precipitation and >15% more Hg concentration that those at Glacier. How to reconcile these disparate spatial differences? Why would one expect simplistic linear extrapolation to capture these differences?

*Our inverse-distance weighted interpolation maps follow procedures developed by the National Atmospheric Deposition Program, which creates CONUS wide maps of annual Hg deposition. Following a similar methodology allows readers to directly compare maps from this analysis to maps produced by NADP. The concentration and deposition maps highlight larger-scale spatial patterns (i.e. North/South, East/West) and will not (and are not intended to) fully capture or model smaller scale depositions trends related to geographic and point/local sourcing. Throughout this section, we clearly mention the limitations of this analysis. As our analysis shows, precipitation is the largest control on annual deposition, and hence the use of the Reanalysis Precipitation product is a valid approach to capture the overall spatial patterns of Hg deposition across this region. We strongly feel that this mapping is very useful to pinpoint the areas of particular concern for Hg deposition and further expansion and investment of monitoring networks.*

Section 3.3 needs to be redone. First the authors conducted a PCA analysis. Usually one uses tracers that represent distinctly different sources, but the metals they used could not seem to do the job. It was not clear why they did a PCA of the metals to begin with. Since Al measurements were not available, the authors decided that Cr and Ni can be used as alternatives of Al, a crustal tracer. Where did they get [xucc]/[Crucc] or [xucc]/[Niucc]? At one point the authors decided that Asian pollution could influence Alaska based on As and Pb (line 489). But there are major anthropogenic sources for Cr and Ni in Asia as well. It was unclear what the authors were trying to do with Figure 6. No interpretation was given but merely description of how the first two components were positively or negatively correlated. What do those correlations really mean?

*We clarified the PCA section. Specifically we identified PC1 as Precipitation with all elements showing a negative correlation due to washout effects. Hg's relationship to PC1 (precipitation) was slightly different from the other metals due to its reactive nature and susceptibility to gaseous reemission.*

This manuscript can be shortened significantly, by removing tutorial material, redundancy, repetition, and passages that merely pointed out the obvious. To be specific, Section 2 can be cut down quite a bit by removing the tutorial stuff in the statistics and trajectory sections.
*We significantly shortened Section 2.2; however, key elements related to the data treatment and statistical handling were left as we feel it is important to highlight data processing (such as treatment of outliers and below detection limit values). Section 2.3 was also shortened to include only key information related to the HYSPLIT model and data processing. Finally, we removed the discussion of backtrajetory analysis for precipitation as well as suggested above.*

In their results and discussion sections they often stated the obvious.
*We edited the manuscript and tried to remove non-critical text and "obvious" discussion related to the interpretation of our results.*

---

## Author Response (AR2)

**Thank you for the review and encouraging ratings. Please find specific responses to Reviewer Comments below in *italics*. We appreciate the thorough review and look forward to follow up research and discussions related to our work.**

The authors have addressed most of my and the other Reviewer's comments. This paper s more readable and the altered discussion/conclusions make it stronger.

A few small comments:
31 and elsewhere:
Use of "in-between". I do not agree that in all instances it should be changed to "in-between"
From:
https://english.stackexchange.com/questions/142168/what-is-the-difference-between-in-between-and-in-between/142172
"According to Merriam-Webster, 'in-between' is used as a noun or adjective whereas 'in between' is an adverb or preposition."

*Throughout the paper (specifically on Line 29) we have addressed the usage of "in-between" and "in between" and consistently follow the Merriam-Webster usage rules above.*

366-372: Your addressal of "dry deposition" strengthens the paper. Are there other studies showing dry deposition of Hg to Alaskan vegetation, soil, or snowpack or just the Obrist et al work? This has been well established at numerous lower latitude sites (as you state).

*An additional citation/discussion (Jaeglé (2010)) of AK specific dry deposition results was added to emphasize the importance of understanding the contributions of both wet and dry deposition.*

*Line 307:*

*"Deposition modelling by Jaeglé (2010) showed that the relative contributions of wet and dry deposition varied spatially throughout Alaska, with wet deposition accounting for approximately 28 percent on average. In agreement, a recent study in the northern Tundra (Obrist, 2017) showed that wet deposition accounted for a minor fraction of overall Hg deposition, with the largest source to the tundra deriving from atmospheric gaseous Hg deposition cycled through arctic vegetation."*

606-615: This makes the paper stronger.

Figure 2: Again I am curious what is wet versus dry deposition at the sites. In April at the lower latitude sites it is likely raining while at the higher latitude ones it is snowing. I am not sure parsing the data this way is reliable for cross comparison. Maybe bin by > or < C at a given site when precipitation falls?

*To be clear, the State of AK/NADP datasets only include wet deposition data. We agree that precipitation phase (i.e. snow vs. rain vs. ice, etc.) causes different atmospheric scavenging processes that no doubt contribute to concentration differences; however, a full breakdown of precipitation phase is not possible with the current dataset. We analyze wet deposition as any precipitation-based (snow, rain, drizzle,*

*sleet) flux. The consistent seasonality of concentrations at all sites suggests that larger-scale transport and precipitation magnitude mechanisms are driving the underlying temporal patterns.*

[revised manuscript text omitted]